# Learning Math Reasoning from Self-Sampled Correct and Partially-Correct Solutions

**Ansong Ni**[1*]  **Jeevana Priya Inala**[2]  **Chenglong Wang**[2]
**Oleksandr Polozov**[3†]  **Christopher Meek**[4‡]  **Dragomir Radev**[1]  **Jianfeng Gao**[2]
[1]Yale University, [2]Microsoft Research, [3]Google, [4]University of Washington
`ansong.ni@yale.edu, {jinala, chenwang}@microsoft.com`

## Abstract

Pretrained language models have shown superior performance on many natural language processing tasks, yet they still struggle at multi-step formal reasoning tasks like grade school math problems. One key challenge of finetuning them to solve such math reasoning problems is that many existing datasets only contain one reference solution for each problem, despite the fact that there are often alternative solutions resembling different reasoning paths to the final answer. This way, the finetuned models are biased towards the limited reference solutions, which limits their generalization to unseen examples. To mitigate this issue, we propose to let the model perform sampling during training and learn from both self-sampled fully-correct solutions, which yield the correct answer upon execution, and partially-correct solutions, whose intermediate state matches an intermediate state of a known correct solution. We show that our use of self-sampled correct and partially-correct solutions can benefit learning and help guide the sampling process, leading to more efficient exploration of the solution space. Additionally, we explore various training objectives to support learning from multiple solutions per example and find they greatly affect the performance. Experiments on two math reasoning datasets show the effectiveness of our method compared to learning from a single reference solution with MLE, where we improve PASS@100 from 35.5% to 44.5% for GSM8K, and 27.6% to 36.2% PASS@80 for MathQA. Such improvements are also consistent across different model sizes. Our code is available at `https://github.com/microsoft/TraceCodegen`.

## 1 Introduction

Recent progress on pretrained language models shows that they are able to achieve human-level performance on various natural language processing tasks with finetuning(Devlin et al., 2019; Brown et al., 2020; Raffel et al., 2020). However, such models still lack the ability to perform multi-step math reasoning even for problems that are intended for grade-school students (Cobbe et al., 2021). Current methods for solving math problems typically rely on generating solutions (a sequence of computation steps) and executing them to obtain the final answer (Cobbe et al., 2021; Austin et al., 2021; Chen et al., 2021a; Chowdhery et al., 2022), as directly generating the final answer would require computational abilities that even the largest models do not possess (Brown et al., 2020; Chowdhery et al., 2022).

When finetuning such models on math reasoning, existing methods often rely on the MLE objective that aims to maximize the log-likelihood of the reference solution for each natural language input. However, in addition to the reference solution, there are often multiple correct solutions for each question, resembling alternative reasoning paths to the final answer. However, those alternative solutions are unseen during training, and this results in model overfitting: the model becomes overly confident in its predictions because it sees the same solution over multiple epochs of training (Bunel et al., 2018; Austin et al., 2021; Cobbe et al., 2021). This leads to poor generalization on unseen

---

*Majority of the work done during an internship at Microsoft Research. †Work started while at Microsoft Research, paper contribution limited to proof-reading. ‡Work initiated while at Microsoft.

inputs and is reflected by the low PASS@$k$ performance, where the model is unable to predict the right answer even when allowed multiple attempts per question.

To mitigate this issue, we propose learning from self-sampled solutions. Concretely, during training time, the model samples alternative solutions, and keeps track of all solutions that are semantically correct with respect to the gold execution result, and learns from all of these correct solutions as opposed to only from the reference. To further improve the effectiveness of learning from self-sampled solutions, we allow the model to learn from partially-correct solutions, whose intermediate states are consistent with intermediate states of known correct solutions. This new technique allows the model to maximally utilize the self-sampling and more efficiently explore the solution space. We also study various common loss functions for learning from multiple targets for a single natural language input, including augmented-MLE, Maximize Marginal Likelihood (MML) and $\beta$-smoothed MML (Guu et al., 2017) and find that their different gradient equations greatly affect the learning capabilities of the model.

We perform experiments on two math reasoning tasks, namely MathQA-Python (Austin et al., 2021) and Grade-School-Math (GSM) (Cobbe et al., 2021), and finetune GPT-Neo models (Black et al., 2021) to generate Python program as solutions from the problem description in natural language. Results show that learning from self-sampled solutions can improve the PASS@100 from 35.5% to 44.5% for GSM, and 27.6% to 36.2% for PASS@80 on a filtered version of MathQA-Python.[1] Moreover, we find that learning from partially-correct solutions generally improves performance over learning from just fully-correct solutions (*e.g.,* +3.0% PASS@100 for GSM8K) as it guides the sampling process, discovering more alternative solutions for learning. Such performance boosts from our proposed methods are also consistent for different model sizes. Ablation on different loss functions shows that MLE-Aug loss is the most effective in learning from multiple targets and yields the most improvements over MLE loss.

## 2 OVERVIEW

**Problem formulation.** We consider the task of generating solutions from math problem descriptions in natural language (NL). Given an NL input $x \in \mathcal{X}$ and the executor $\mathcal{E} : \mathcal{Y} \to \mathcal{Z}$, the goal is to generate a solution $y \in \mathcal{Y}$ that executes to the expected answer $z^* \in \mathcal{Z}$, *i.e.,* $\mathcal{E}(y) = z^*$.

**Standard approach and its limitation.** The standard approach is to assume that we have a dataset of paired NL input $x$ and reference solution $y^*$. Most datasets typically only provide one reference solution for a particular NL input. Then, a parameterized model $P_\theta$ is learned with the *Maximum Likelihood Estimation* (MLE) objective from the NL-Solution pair $(x, y^*)$ as:

$$\mathcal{L}_{\text{MLE}}(x, y^*, P_\theta) = -\log P_\theta(y^*|x) \tag{1}$$

The builtin assumption of using Eq. 1 for learning is that only the reference solution $y^*$ is correct. However, this assumption is clearly untrue for the math reasoning problem as typically multiple reasoning paths can achieve the correct final result. With only one reference solution as target for learning, Eq. 1 would encourage the model to put all probability mass on $y^*$, which could easily lead to *overfitting* (Bunel et al., 2018; Austin et al., 2021; Cobbe et al., 2021).

**Overview of our approach.** While manually collecting additional reference solutions for each specification is a laborious process (Austin et al., 2021; Cobbe et al., 2021; Schuster et al., 2021), in our work, we explore an alternate approach: where the model self-samples additional correct (or partially-correct) solutions and learns from them during training. Fig. 1 shows an example: for the question $x$, our model was able to self-sample an alternative solution $\hat{y}$ that is different from the reference solution $y^*$ provided in the dataset. Looking at the intermediate states shown on the right, we can see that both these solutions execute to produce the sample desired output, *i.e.,* $\hat{z} = z^*$, as noted with solid red boxes. Taking this one step further, our approach can also identify partially-correct solutions from its samples. For example, on the bottom left, we show a sampled solution $\hat{y}'$ that is incorrect only because of an error in its last two steps. But we identify a prefix $\hat{y}'_{\leq 5}$ of it as partially-correct because the intermediate state $\hat{s}'_5$ for this prefix matches the intermediate state $s^*_5$ of a known correct solution $y^*$ (noted as dashed red boxes) and yet syntactically different from $y^*$. Based on these observations and intuitions, we introduce our approach in the following sections.

---

[1]We choose different $k$ for evaluating PASS@$k$ to be consistent with previous work.

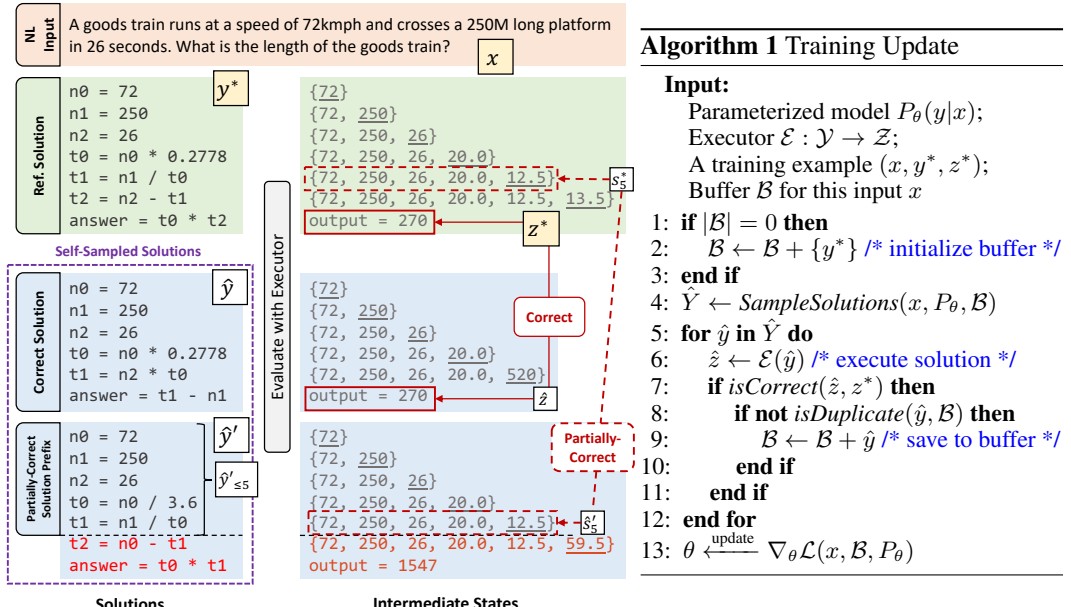

Figure 1: Examples of self-sampled correct and partially-correct solutions from MathQA (more in Appendix D). The steps and intermediate states marked in red are *incorrect*.

# 3   LEARNING FROM SELF-SAMPLED SOLUTIONS

We now formally present our approach. There are three main steps: 1) *sampling* 2) *filtering* and 3) *learning* as shown in Alg. 1. Here we mainly introduce the self-sampling framework using only fully-correct solutions and the extensions with partially-correct solutions will be introduced in § 3.3.

## 3.1   ONLINE SAMPLING AND FILTERING

For each specification $x$, we maintain a buffer $\mathcal{B}$ to save the different solutions that are correct, *i.e.,* evaluate to the correct result. Note that the buffers are persistent and cumulative across training epochs. To add more solutions in $\mathcal{B}$, we perform online sampling and filtering as follows.

**Online sampling (line 4 in Alg. 1):** With the NL question $x$ from each example $(x, y^*, z^*)$ as input, the model samples a set candidate solutions $\hat{Y} = \{\hat{y}_i\}_{i=1}^n \sim P_\theta(\hat{y}|x)$;

**Filtering incorrect solutions(line 7 in Alg. 1):** As not all sampled solutions in $\hat{Y}$ are correct (thus not suitable for learning), we filter out all incorrect solutions in $\hat{Y}$, *i.e.,* $\hat{Y}^* = \{\hat{y}|\hat{y} \in \hat{Y}; \mathcal{E}(\hat{y}) = z^*\}$;

**Filtering duplicate solutions (line 8 in Alg. 1):** Because the model can sample solutions that are correct but are "trivial variants" of other already saved solutions (*e.g.,* the solution differs from another solution only in white spaces, comments or trivial steps like "x = x * 1.0"), we further filter the buffer to remove them. This is essential as all saved solutions will be directly used for learning and such undesired behavior from the model will be encouraged without the filtering process.[2] Concretely, we first perform filtering based on the linearized abstract syntax trees (ASTs) to eliminate the differences in white space, etc; then we set a constraint on maximum number of lines using the number of lines in $y^*$ as the reference to prevent saving solutions with trivial steps.

## 3.2   LEARNING FROM MULTIPLE TARGETS

With self-sampling, each natural language question is paired with multiple solutions as targets for learning. Here we discuss some common loss functions for the multi-target learning problem, with

---

[2]Our preliminary experiments also show that the performance greatly degenerates when such trivial variants are left in the buffer for learning.

| Name | Loss Functions $\mathcal{L}(x, \mathcal{B}, P_\theta)$ | Gradients $\nabla_\theta(x, \mathcal{B}, P_\theta)$ |
|---|---|---|
| MLE | $-\log P_\theta(y^*\|x)$ | $-\nabla_\theta \log P_\theta(y^*\|x)$ |
| MLE-Aug | $-\sum_{\hat{y}\in\mathcal{B}} \log P_\theta(\hat{y}\|x)$ | $-\sum_{\hat{y}\in\mathcal{B}} \nabla_\theta \log P_\theta(\hat{y}\|x)$ |
| MML | $-\log \sum_{\hat{y}\in\mathcal{B}} P_\theta(\hat{y}\|x)$ | $-\sum_{\hat{y}\in\mathcal{B}} \frac{P_\theta(\hat{y}\|x)}{\sum_{\tilde{y}\in\mathcal{B}} P_\theta(\tilde{y}\|x)} \nabla_\theta \log P_\theta(\hat{y}\|x)$ |
| $\beta$-MML | $-\frac{1}{\beta}\log \sum_y P_\theta(\hat{y}\|x)^\beta$ | $-\sum_{\hat{y}\in\mathcal{B}} \frac{P_\theta(\hat{y}\|x)^\beta}{\sum_{\tilde{y}\in\mathcal{B}} P_\theta(\tilde{y}\|x)^\beta} \nabla_\theta \log P_\theta(\hat{y}\|x)$ |

Table 1: Comparison of loss functions and their gradients over multiple reference $\mathcal{B}$. Note that they all degenerates to MLE when only the gold reference solution is used as target, *i.e.,* $\mathcal{B} = \{y^*\}$.

a focus on how each target contributes to the gradient. The loss functions and their gradients are shown in Tab. 1.

**Augmented MLE (MLE-Aug):** This objective augments MLE with multiple targets simply by summing the loss from multiple solutions in $\mathcal{B}$, which is equivalent as minimizing the KL-divergence from $P_\theta(y|x)$ to $Q(y|x) = \frac{1}{|\mathcal{B}|} \cdot \mathbb{1}_\mathcal{B}(y)$, where $\mathbb{1}(\cdot)$ is a set indicator function. It encourages the model to put equal weights on all targets by ensuring that all targets equally contribute to the gradient.

**Maximum Marginal Likelihood (MML):** MML attempts to approximate $P_\theta(z^*|x)$ by marginalizing over the correct solutions in $\mathcal{B}$. However, for each target $\hat{y} \in \mathcal{B}$, the gradient of it is in proportion to the likelihood $P_\theta(\hat{y}|x)$ given by the model, which results in a positive feedback loop during gradient updates. It encourages the model to still put a majority of the probability on one of the solutions in $\mathcal{B}$ as noted in (Guu et al., 2017).

**$\beta$-smoothed MML ($\beta$-MML):** Proposed in Guu et al. (2017), the $\beta$-MML objective is an extension of MML with a hyperparameter $\beta \in (0, 1]$ to adjust weights of the gradient from each target. It an interpolation between MML and MLE-Aug objectives, more specifically, it recovers MML when $\beta = 1$ and its gradient is equivalent to that of MLE-Aug when $\beta \to 0$.

Empirically, we find that these distinctions between those loss functions greatly affects the model performance (Fig. 3), especially when partially-correct solutions are included for learning.

## 3.3 LEARNING FROM PARTIALLY-CORRECT SOLUTIONS

Besides learning from self-sampled *fully-correct solutions* (FCSs), we can also let the model learn from *partially-correct solutions* (PCSs). Our motivation is that the model often encounter solutions that are close to being correct as they only make mistakes in the last few steps (*e.g.,* Fig. 1), and these partially-correct solutions provide additional learning opportunities. Learning from PCSs could also address the issue that the sampler may have a low chance of encountering fully-correct solutions for complex tasks due to the sparse solution space.

### 3.3.1 IDENTIFYING PARTIALLY-CORRECT SOLUTIONS

When the model samples a solution that does not produce the desired answer, we want to identify if a prefix of this solution is partially correct, *i.e.,* it performs some of the necessary computation steps needed for the correct solution, so that the model can additionally learn from these potentially unseen prefixes in the next iteration. A challenge here is figuring out when a prefix is partially correct. Ideally, we want to say a prefix $y_{\leq i}$ is partially correct if there exists a suffix $y_{>i}$ such that their concatenation $(y_{\leq i}||y_{>i})$ is a correct solution. There are two caveats here: (1) if there is no length restriction on the suffix, it is always possible to find a suffix that complements any prefix (*e.g.,* a full gold solution is one such suffix); and (2) it is computationally very expensive to search for all suffixes (even with a length restriction) to check if a prefix can be completed to a correct solution.

To overcome these challenges, we leverage the gold reference solutions and any self-sampled fully-correct or even partially-correct solutions to help identify new partially-correct prefixes. The idea is to identify a prefix as partially correct if it produces a set of intermediate values (upon execution) that exactly matches the set of intermediate values produced by a prefix of a known correct or partially-correct solution. For such a prefix, we know that there exists a reasonable complement suffix based on the suffix of the known solutions. Note that, this definition of partial correctness is conservative compared to the ideal definition above, but it makes the computation significantly tractable.

---

**Algorithm 2** *SampleSolutions*$(x, P_\theta, \mathcal{B})$ with partially-correct solutions

---

**Input:** Model $P_\theta(y|x)$; the NL input $x$ and a set of partially-correct solutions $\mathcal{B}$
**Output:** Solution samples $\hat{Y}$.

1: Select $\hat{y}_{\leq i} \in \mathcal{B} \setminus \{\hat{y} | \mathcal{E}(\hat{y}) = z^*\}$ uniformly at random /* sample PCS prefix for completion */
2: Sample a set of completions $Y_p \sim P_\theta(\hat{y}_{>i} | \hat{y}_{\leq i}, x)$
3: $\hat{Y} \leftarrow \{[\hat{y}_{\leq i} || \hat{y}_{>i}]\}_{\hat{y}_{>i} \in Y_p}$ /* concatenate completions with the solution prefix */
4: **return** $\hat{Y}$

---

Below, we formally define this notion of partial solutions that leverages existing known fully and partially correct solutions.

**Intermediate state.** Given a solution $y = (u_1, ..., u_t)$ where $u_i$ is the $i$-th reasoning step, we define the intermediate state $s_i$ as the set of all variables values in the scope after executing the first $i$ steps $y_{\leq i} = (u_1, ..., u_i)$, which we call a *prefix* of this solution. It is easy to see that the prefixes $y_{\leq i}$ and intermediate states $s_i$ of a solution construct a bijective function, which is also illustrated in Fig. 1. Note that the state representation is name-agnostic since variable names do not typically contributes to the semantics of the solutions.

**State-based equivalence and partial correctness.** Given the definition of the intermediate state, we say the prefixes of two solutions, $y_{\leq i}$ and $y'_{\leq j}$, are *semantically equivalent* if and only if $s_i = s'_j$, *i.e.,* those two solutions produces the exact same set of variable values. And then we define *partial correctness* as follows: a solution prefix $y_{\leq i}$ is partially-correct if and only if it is semantically equivalent to the prefix of another known partially-correct solution $y^*_{\leq j}$. As we keep all known partially-correct solutions in the buffer $\mathcal{B}$, formally:

$$PartiallyCorrect(y_{\leq i}) \iff \exists y^* \in \mathcal{B}. \ \exists j \leq |y^*| \ s.t. \ s^*_j = s_i$$

### 3.3.2 MODIFICATIONS TO THE MAIN ALGORITHM

To support learning from partial solutions, we modify Alg. 1 as follows to enable buffering and sampling from partial solutions. The fully updated algorithm is shown in Appendix C.

**Guided-Sampling:** In § 3.1, we mentioned that full solutions are sampled for each question $x$ as $\hat{y} \sim P_\theta(\hat{y}|x)$. With PCS prefixes, compared with sampling a solution from scratch, generating solutions with these prefixes reduces the generation length thus the model can more efficiently explore the solution space. This guided sampling process is described in more detail in Alg. 2. Note that since the empty solution $y^0$ is in the buffer $\mathcal{B}$ since initialization, therefore model can still generate and explore the space from scratch and not always follows the existing solution prefixes.

**Identify partially-correct prefixes:** As mentioned in § 3.3, if a solution $\hat{y}$ does not produce the expected result $z^*$ but its prefix $\hat{y}_{\leq i}$ is partially-correct, the model can still learn from its prefix. However, an important task here is to identify the longest partially-correct prefix for learning, in other words, locate the exact step that the solution deviates from a correct reasoning path. We can achieve this simply by backtracking the intermediate states and find the first state that is equivalent to any of the states from a saved solution. [3]

**Filtering solution prefixes:** With the inclusion of partially-correct solutions, we need to slightly change the two filtering criteria in § 3.1. For deduplication, while we still use AST to rule out changes with non-semantic tokens such as white space, we also check if the partially-correct solution prefix $\hat{y}_{\leq i}$ is a prefix of another known PCS in $\mathcal{B}$. For the same reason, when saving a new partially-correct solution $\hat{y}$, we need to prune out any existing solution in $\mathcal{B}$ that is a prefix of $\hat{y}$. As for the length constraint, the same principle still applies, but now it is compared against other partially-correct solution that *executes to the same state*.

**Learning objective:** As partially-correct solutions are solution prefixes $y_{\leq i}$ missing the later part $y_{>i}$, with an auto-regressive generation model, the learning of $P_\theta(y_{\leq i}|x)$ is independent of $y_{>i}$. Thus the learning objectives in § 3.2 do not need to change with the inclusion of PCS in the buffer for learning. The only difference is that the end-of-sequence "⟨eos⟩" token is not appended to the PCS as those solutions are not yet finished.

---

[3] In practice, we use a `state → solution prefix` dictionary and the lookup takes a negligible amount of time.

## 4 EXPERIMENTS

### 4.1 EXPERIMENTAL SETUP

**Datasets.** We evaluate on two math reasoning datasets, in which we generate straight-line Python programs as solutions to solve math problems described in natural language. We finetune the language models to output such program solutions using only the natural language problem description as the input.

▷ **MathQA-Python-Filtered:** The original MathQA-Python consists of 19.2K training examples of NL and Python program pairs (Austin et al., 2021). However, we find the raw dataset to contain many questions that share the same question templates and only differ in concrete number across the train/dev/test sets. To better understand the generalization of the trained models, we derive a deduplicated version of the dataset by first merging the train and dev data and then perform template-based deduplication. Partly inspired by Finegan-Dollak et al. (2018), we re-split the train and dev set based on the question templates, resulting in 6.8K/0.7K train/dev data for the filtered version.[4] While we mainly experiment on the filtered version, we report performance on both versions when compared with previous methods.

▷ **GSM5.5K-Python:** The grade-school-math (GSM8K) dataset (Cobbe et al., 2021) contains 7.5K training data points. Since it only provides natural language solutions with math formulas and does not have a dev set, we first reserved 20% of the training data as dev set, then automatically converted the formulas to program solutions in the same style as MathQA-Python. As the result, we finetune our models with the 5.5K successfully converted training examples. Note that the natural language solutions/explanations are not used as input to the models in our experiments.

**Evaluation metrics:** Following recent work in neural program synthesis (Austin et al., 2021; Chen et al., 2021a; Chowdhery et al., 2022) and math reasoning (Cobbe et al., 2021), we use PASS@$k$ as our main evaluation metric. It allows the model to sample $k$ solutions for each question and the task is considered solved if any one of the $k$ solutions is correct, so PASS@$k$ can also be seen as the fraction of problems in the test/dev set being solved given $k$ attempts. More details (*e.g.,* temperature) can be found in Appendix A.

**Model training:** We use GPT-Neo (Black et al., 2021) as our language model and mainly study two model sizes, 125M and 2.7B.[5] Following previous work (Austin et al., 2021), we evaluate all PASS@$k$ on the same model checkpoint that has the best PASS@1 score, but note that it might not be the best checkpoint for other $k$ values (more discussion in Appendix E). Detailed hyperparameter settings can also be found in Appendix A.

### 4.2 MAIN RESULTS

**Learning from self-sampled solutions improves PASS@$k$.** Fig. 2 shows the performance on the two datasets by learning from self-sampled FCSs and PCSs using MLE-Aug (orange bars), compared with MLE on single reference solution (blue bars). We can see that our proposed method can greatly improve PASS@$k$, especially for higher $k$ values. By comparing different model sizes, we can see that learning from self-sampled solutions can help with both small and large models, with a +12.3% and +9.0% PASS@100 improvement on GSM5.5K-Python for GPT-Neo-125M and GPT-Neo-2.7B, respectively and a +3.1% and +8.6% PASS@80 improvement on MathQA-Python-Filtered for GPT-Neo-125M and GPT-Neo-2.7B, respectively. We note that our approach does not improve PASS@1, which is expected as learning from multiple targets mainly helps with increasing the diversity of the sampled solutions rather than improving the most-probable solution (for which MLE is better suited).

**Partially-correct solutions improve model performance.** We next show the effects of including partially-correct solutions on PASS@$k$ performance in Fig. 2 (green bars vs orange bars) and the number of saved FCSs and PCSs in Fig. 4. First, we observe from Fig. 4 that using partial correctness not only results in PCSs being saved and directly learned from, but it also boosts the number of FCSs being found with the guided-sampling process. As a result, most PASS@$k$ performances drop if we

---

[4]We will release the processing scripts for replication and comparison.

[5]We choose GPT-Neo because it was the only public language model that have been pretrained on code when we conduct the experiments.

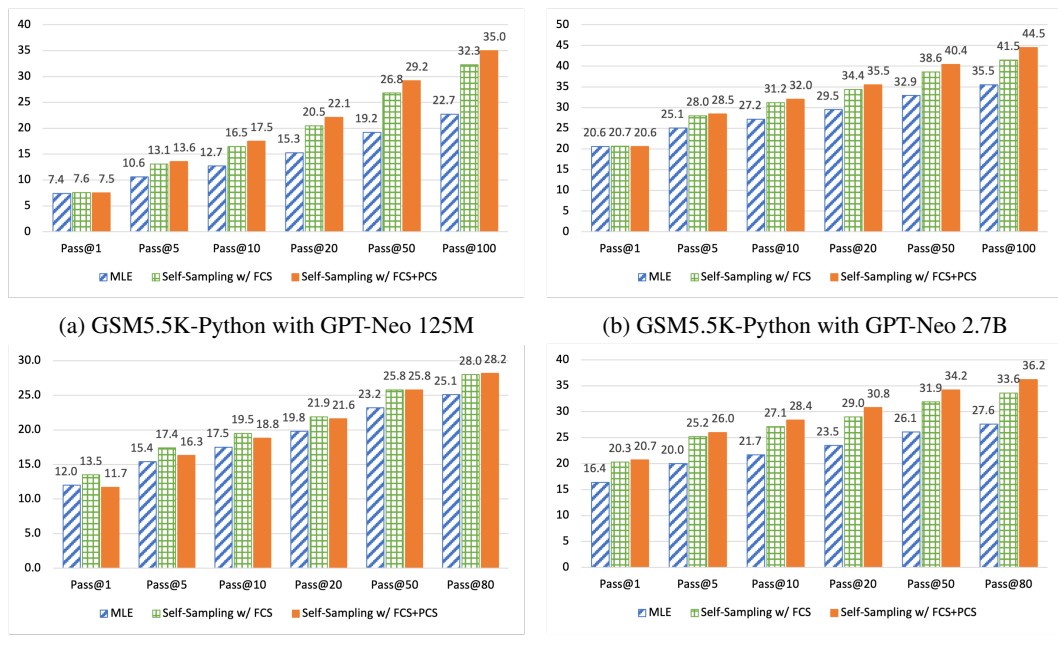

(a) GSM5.5K-Python with GPT-Neo 125M

(b) GSM5.5K-Python with GPT-Neo 2.7B

(c) MathQA-Python-Filtered with GPT-Neo 125M

(d) MathQA-Python-Filtered with GPT-Neo 2.7B

Figure 2: Percentage of the problems solved (PASS@$k$) on the dev set of GSM5.5K-Python and MathQA-Python-Filtered, comparing our self-sampling approach and the common MLE objective. All our methods include partially-correct solutions and use the MLE-Aug loss for learning.

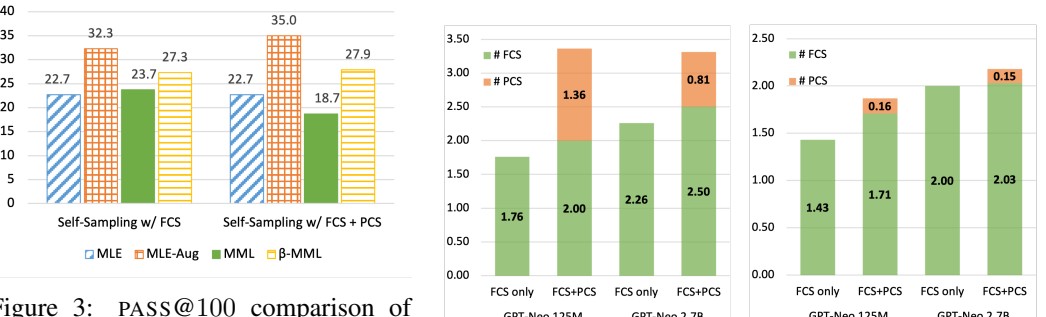

Figure 3: PASS@100 comparison of various loss functions (§ 3.2) under different self-sampling strategies. Results are on the dev set of GSM5.5K-Python with finetuned GPT-Neo 125M model. $\beta = 0.25$ for $\beta$-MML. Full results available as Tab. 5 in Appendix B.

Figure 4: Number of saved FCSs and PCSs per problem for GSM5.5K-Python (left) and MathQA-Python-Filtered (right), with different self-sampling strategies and model sizes. # FCSs *includes* the reference solution.

do not include partially-correct solutions in the buffer, as the model learns from a smaller number of FCSs and PCSs as targets. The one exception is the GPT-Neo 125M model on the MathQA-Python-Filtered dataset, where we do not observe any advantage/disadvantage of using PCSs.

**MLE-Aug loss function works the best.** We next study the effects of different objective functions for learning from multiple targets as described in § 3.2. We also experiment under different self-sampling strategies (*i.e.,* FCS only or FCS + PCS), and our experiment results on GSM5.5K-Python with the GPT-Neo 125M model are shown in Tab. 5. We can see that MLE-Aug loss results in the biggest improvement compared to other losses both with just FCSs and with FCSs + PCSs. MML performs the worst: it only marginally improves over MLE with only FCS and performs worse than MLE when also learning from PCSs. As discussed in § 3.2 and Tab. 1, the gradient of MML is in proportional to the likelihood given by the model, thus it encourages the model to put all weight on one solution in the buffer. As MLE already learns from the gold reference solution, it is hard for

| Models | Original Version | | Filtered Version | |
|---|---|---|---|---|
| | PASS@1 | PASS@80 | PASS@1 | PASS@80 |
| *Previous work*: | | | | |
| Codex Davinci[†] (Chen et al., 2021a) | 6.0 | 42.0 | 5.0 | **40.0** |
| LaMDA 68B[*] (Austin et al., 2021) | - | 79.5 | - | - |
| LaMDA 137B[*] (Austin et al., 2021) | - | 81.2 | - | - |
| *Ours:* | | | | |
| GPT-Neo 125M w/ self-sampling FCS + PCS | **77.6** | **84.7** | 11.7 | 28.2 |
| GPT-Neo 2.7B w/ self-sampling FCS + PCS | - | - | **20.7** | 36.2 |

Table 2: Comparison with previous methods on the original (test set used) and filtered version (dev set used) of the MathQA-Python dataset. [*]: model not pretrained on code. [†]: few-shot learning results. -: no results available.

MML to make improvements with self-sampled solutions, and the performance may even decrease when MML puts all weight on an incomplete partially-correct solution. In contrast, the gradients of MLE-Aug objective are equally distributed among the targets, which leads to more diversity in its generation due to a more balanced source of learning signals. $\beta$-MML loss is proposed to alleviate the aforementioned issue for MML loss, but we do not observe an advantage of using it instead of the MLE-Aug loss in our experiments.

### 4.3 Additional Analysis

**Diversity of the solutions.** By inspecting the $k$ generated solutions for each task, we find that there is more diversity in the solutions that the model generates using our method. More specifically, we calculate the ratio of *unique* solutions from the 100 samples for the comparison in Fig. 2a, and find that 30.5% of them are unique for our approach but only 20.8% for the model trained with MLE.

**Dynamics between # of PCSs and FCSs saved in the buffer.** As discussed above, more saved solutions typically results in better PASS@$k$ performance. Interestingly, when comparing different model sizes, we can see that while the sum of partially and fully-correct solutions sampled and saved in the buffer are about the same (*i.e.,* 3.36 and 3.31) for GSM5.5K-Python dataset in Fig. 4, around 60% of them are FCS for the small model while it is 76% for the larger model. The difference in percentage of PCSs left in the buffer also reflects the model's ability for completing partially-correct solution prefixes. We also find that during early stages of training, the number of PCSs rapidly grows while the model is relatively weak to sample FCSs, thus the PCSs help enriching the learning signal and preventing overfitting early-on. More discussions about this can be found in Appendix E.

**Comparison to previous works** Here we compare with previous work on both the original and the filtered versions of MathQA-Python datasets in Tab. 2. On the original dataset, self-sampling with GPT-Neo 125M is able to outperform previous methods that finetune 137B model pretrained on natural language. We also compare with Codex model used in a few-shot setting (more details in Appendix A), and find that on the much harder filtered dataset, a 2.7B GPT-Neo model finetuned with our methods obtains much better PASS@1 but with lower PASS@80. By inspecting the output from Codex, we discover that its outputs are much more diverse than finetuned models, which contributes to a higher PASS@80 even under the few-shot setting. Comparison with previous work on the GSM dataset is in Appendix B due to limited space.

## 5 Limitations and Future Work

**More general definition of (partial) correctness.** In this work, we define partial correctness based on state-based solution equivalence. This is a conservative way for defining solution equivalence as it requires exact match of the sets of variable values, but a solution could be partially correct and yet, not have an exact match of variable values because some of these values may not needed for future computation. In the future, we want to explore ways to relax this restriction that will help us find more partially correct solutions in an efficient manner. Besides, our partial correctness definition requires the existence of at least one fully-correct solution and when such reference solution is

not available from the dataset (*i.e.,* in a weakly-supervised setting), we would need to first sample an FCS that matches the gold execution result to begin with. In addition, we simply use the matching of execution results to define correctness, which is susceptible to spurious solutions that achieves the correct result by coincidence. For math reasoning, we find such spurious solutions to be quite rare[6], as the correct answer is typically numeric which is less likely for a semantically wrong solution to obtain the correct answer by chance. But methods as Zhong et al. (2020); Chen et al. (2022) may be needed for this definition of correctness to be more robust on other domains.

**Towards generating general programs.** While we focus on the domain of generating solutions for math reasoning in this work, here we reflect on how our method can be applied to program synthesis in general. However, general programs might contain complex structures such as conditions (*e.g.,* `if-else`) or loops (*e.g.,* `while-do`) as opposed to straight-line programs in the math-reasoning domain. Dealing with these complex structures poses additional challenges because most neural program synthesis models perform left-to-right auto-regressive generation, and the changes to the control flow break the alignment between program generation and program execution (Chen et al., 2018; 2021b; Nye et al., 2021). There are two potential ways to extend our technique to address the problem. First, we can treat a branch or a loop as an atomic unit (*i.e.,* a block whose state is the state after executing all statements within it), then we can apply state-based equivalence in the same way. Second, because our technique only requires execution after the full programs are generated, we can still evaluate and compare program states based on intermediate states.

## 6 RELATED WORK

**Weakly-supervised semantic parsing.** Many previous work in learning semantic parsers from weak supervision follows the same process of sampling programs and maximizing the probability of the correct ones (Krishnamurthy et al., 2017; Guu et al., 2017; Min et al., 2019; Ni et al., 2020). Our work differs as our tasks contain one reference solution for each task as opposed to only the final answer like weakly-supervised semantic parsing tasks. Thus, our work leverages the reference solution for sampling and defines partial correctness based on known solutions. Because of the problem setup difference, we found that the conclusions in Guu et al. (2017) about loss functions do not generalize to our case.

**Execution-guided code generation.** Our work relates to execution-guided code generation as we leverage intermediate states of math solutions to guide the sampling process. In code generation literature, intermediate program execution states are used to prune the search space (Liang et al., 2017; Wang et al., 2018; Li et al., 2022) or condition further generation on the execution states(Chen et al., 2018; Ellis et al., 2019; Nye et al., 2020; Chen et al., 2021b; Nye et al., 2021). The key difference of these methods from ours is that they require doing both decoding and execution at inference time, while our work only uses execution during training, which reduces decoding overhead.

**Learning from partial reward for program synthesis.** There are parallels between multi-target learning and the reinforcement learning setting with sparse rewards for generating programs (Liang et al., 2017; 2018; Simmons-Edler et al., 2018; Bunel et al., 2018; Agarwal et al., 2019). Similarly, our approach of identifying partial correctness of solutions is similar to partial rewards. But instead of discounting an entire trajectory with a low reward as in RL, we truncate the solution to a partially-correct prefix and assign it the "full reward", which is a main contribution of this work.

## 7 CONCLUSION

We propose to let pretrained language models sample additional solutions for each problem and learn from the self-sampled solutions that are correct or partially-correct. We define partial correctness by tracing and matching intermediate execution states. We experiment on different math reasoning tasks and show that such partially-correct solutions can help more efficient exploration of the solution space and provide useful learning signal, which improves the PASS@$k$ performance. Overall, our proposed method can improve PASS@$k$ from 3.1% to 12.3% compared to learning from a single solution with MLE.

---

[6]We manually inspected the self-sampled FCSs by GPT-Neo 2.7B on 100 tasks of GSM5.5K and found spurious solutions only exist for 3 of them.

ACKNOWLEDGEMENTS

The authors would like to thank Jackson Woodruff, Pengcheng Yin, and the anonymous reviewers for the useful discussion and comments.

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

APPENDIX

## A  EXPERIMENT SETTING DETAILS

| Name | MathQA | GSM8K. |
|---|---|---|
| # Training Steps | 50K | 25K |
| Learning Rate (LR) | | 1.0e-4 |
| Optimizer | | AdamW |
| Adam Betas | | (0.9, 0.999) |
| Adam Eps | | 1.0e-8 |
| Weight Decay | | 0.1 |
| LR Scheduler | | Linear w/ Warmup |
| # LR Warm-up Steps | | 100 |
| Effective Batch Size | | 32 |
| FP Precision | FP 32 for 125M, FP16 for 2.7B | |
| Gradient Clipping | | 1.0 |

Table 3: The hyperparameters used for model training on two different types of datasets.

**Hyperparameters.** All hyperparameters for training is shown in Tab. 3. We use $\beta = 0.25$ in the experiments with $\beta$-MML, as a result of enumeration search among the values of $\{0.1, 0.25, 0.5, 0.9\}$. We use the default AdamW optimizer settings and slightly tuned the learning rate by trying out several values between 1.0e-3 and 1.0e-5. The difference in floating point precision is to fit the GPT-Neo 2.7B model into the memory of the GPUs. All experiments are conducted on V100-32GB GPUs.

**PASS@$k$ evaluation.** We use temperature sampling and sample $n$ solutions with $T = 0.8$, where $n = 80$ for MathQA and $n = 100$ for GSM to evaluate PASS@$n$, to be maximally consistent with previous work (Austin et al., 2021; Cobbe et al., 2021; Chowdhery et al., 2022). We also report PASS@$\{5, 10, 20, 50\}$ using the $n$ samples and the unbiased estimator proposed in Chen et al. (2021a). We use $T = 0.2$ to sample 1 solution per specification and evaluate PASS@$1$.

**Codex few-shot settings.** We estimate the Codex (Chen et al., 2021a) performance under the few-shot settings. More specifically, the prompt consists of a natural language task description "`# Generate Python code to solve the following math word problems:`" and four examples, following previous work (Chowdhery et al., 2022). Each example consists of the NL specification as a one-line comment and the gold program solutions. We evaluate PASS@$k$ for Codex using the same sampling methods as above.

**Details for self-sampling.** During a training step, we sample one solution[7] for each task (*i.e.,* natural language problem) in the batch, *i.e.,* $|\hat{Y}| = 1$ in Alg. 1 and Alg. 2. Thus for each gradient update, we first compute the loss for each task based on the saved solutions in the buffer and loss functions described in Tab. 1, then it is averaged across the 32 tasks in the batch. Note that the total number of samples we generate per task throughout training is also scaled up by the number of training epochs, which is 235 for MathQA-Python-Filtered, 83 for MathQA-Python and 145 for GSM5.5K-Python. For sampling temperature, we use the same setting as inference time, with $T = 0.8$.

## B  ADDITIONAL EXPERIMENT RESULTS

**Comparing GSM performance with previous work.** Here we compare our method with previous work on the original test sets of GSM8K. The results are shown as Tab. 4. On GSM8K, some of the prior works are evaluated on a different format of NL inputs than ours, so they are not directly comparable, but we still include them to help better position the performance of our methods. We test Codex using the same input in a few-shot setting, and we find that similar with the

---

[7]We also experiment with higher sampling budgets but do not observe significant improvements.

| Models | PASS@1 | PASS@100 |
|---|---|---|
| *Previous work*: | | |
| OpenAI 6B[*][♣] (Cobbe et al., 2021) | 21.8 | 70.9 |
| PaLM-Coder 540B[†][♣] (Chowdhery et al., 2022) | **50.9** | - |
| LaMDA 137B[*][†][♣] (Chowdhery et al., 2022) | 7.6 | - |
| Codex Cushman[†] (Chen et al., 2021a) | 5.0 | 58.0 |
| Codex Davinci[†] (Chen et al., 2021a) | 17.0 | **71.0** |
| *Ours*: | | |
| GPT-Neo 2.7B w/ self-sampling FCS + PCS | 19.5 | 41.4 |

Table 4: Compare with previous methods on the original test set of GSM8K dataset. [*]: model not pretrained on code. [†]: few-shot learning results. [♣]: different setting from ours[8].

| Self-Sampling | Loss Func. | # Sols. in $\mathcal{B}$ | | PASS@k(%) | | | | | |
|---|---|---|---|---|---|---|---|---|---|
| | | FCS | PCS | k=1 | k=5 | k=10 | k=20 | k=50 | k=100 |
| - | MLE | - | - | 7.4 | 10.6 | 12.7 | 15.3 | 19.2 | 22.7 |
| FCS only | MML | 1.48 | - | 6.9 | 11.0 | 13.3 | 16.0 | 20.1 | 23.7 |
| | MLE-Aug | **1.76** | - | 7.6 | **13.1** | **16.5** | **20.5** | **26.8** | **32.3** |
| | $\beta$-MML | 1.57 | - | 7.5 | 11.7 | 14.5 | 17.9 | 23.1 | 27.3 |
| FCS + PCS | MML | 1.40 | 1.10 | 5.5 | 9.0 | 11.0 | 13.1 | 16.2 | 18.7 |
| | MLE-Aug | **2.00** | **1.36** | 7.5 | **13.6** | **17.5** | **22.1** | **29.2** | **35.0** |
| | $\beta$-MML | 1.62 | 1.14 | 7.2 | 12.0 | 14.9 | 18.4 | 23.6 | 27.9 |

Table 5: Full comparison of various loss functions (§ 3.2) with different self-sampling strategies. Results are on the dev set of GSM5.5K-Python with GPT-Neo 125M as the base model. Best performance within the same category is in **bold** and ones *worse than MLE* is underlined. $\beta = 0.25$ for $\beta$-MML.

result on MathQA in Tab. 2, our method achieves better PASS@1 while being significantly worse in PASS@100 compared with Codex. We hypothesize that as Codex model is used tested few-shot setting and not finetuned, it does not suffer from the overfitting issue we mentioned. This leads to great diversity but poor accuracy during generation. However, due to the little information we have about Codex (*e.g.,* model size, training data), it is hard to derive any further conclusion.

**Ablation results on loss functions.** Here we show the full results on the ablation of loss functions in Tab. 5. We can see that trends observed from PASS@100 in Fig. 3 are consistent with other PASS@k results, as MLE-Aug loss beats other two loss functions on all PASS@k. And using MML loss when adding PCSs for learning results in worse performance than MLE for PASS@1 as well. Moreover, from the number of FCSs and PCSs saved in the buffer $\mathcal{B}$, we can also observe that using MLE-Aug loss results in more FCSs and PCSs being saved, thus further encourages diversity in generation.

## C  FULL LEARNING ALGORITHM WITH PARTIAL CORRECTNESS

Our general learning framework in shown as Alg. 1 and it is further extended in § 3.3. Here we show a complete version of the algorithm with using partially-correct solutions in Alg. 3. Additionally, here are the detailed explanation of the data structure and functions used in it:

▷ **Mapping** $\mathcal{M}$: This is a data structure that maps an intermediate state to a set of solution (prefixes) that execute to that state, *i.e.,* $\mathcal{M} : \mathcal{S} \rightarrow \mathcal{Y}^n$. In this mapping, we save *all* PCSs and their intermediate states, *including all prefixes* of any PCS. We use this to significantly speed up the lookup process as mentioned in § 3.3.2;

▷ **Function** *PartialCorrectnessCriteria*$(s_i, \mathcal{M})$: Since all states for all known PCSs are saved in $\mathcal{M}$, to know whether a prefix $\hat{y}_{\leq i}$ is partially-correct, we only need to check if its state matches any

---

[8]Natural language explanations of the solutions are used as input and the few-shot exemplars are not in the same format as ours.

---

**Algorithm 3** Training Update with Partially Correctness

---

**Initialize: (only once before training starts)**
    Solutions buffer $\mathcal{B} = \{y^0, y^*\}$ with an empty and the reference solution
    Reference solution states $(s_1^*, s_2^*, ..., s_t^*)$ where $s_i^* = \mathcal{T}(y_{\leq i})$
    State-prefixes mapping $\mathcal{M} = \{s_i^* \to \{y_{\leq i}\}\}_{i=1}^t$
**Input:**
    Parameterized model $P_\theta(y|x)$
    A training example $(x, y^*, z^*)$
    Tracing function $\mathcal{T} : \mathcal{Y} \to \mathcal{S}$ to obtain intermediate states
1:  $\hat{Y} \leftarrow SampleSolutions(x, P_\theta, \mathcal{B})$ /* call Alg. 2 */
2: **for** $\hat{y}$ **in** $\hat{Y}$ **do**
3:     **for** $i \leftarrow |\hat{y}|; i \neq 0; i \leftarrow i - 1$ **do**
4:         $s_i \leftarrow \mathcal{T}(\hat{y}_{\leq i})$ /* get intermediate state for each solution prefix $\hat{y}_{\leq i}$ */
5:         **if** $PartialCorrectnessCriteria(s_i, \mathcal{M})$ **then**
6:             $Y_S \leftarrow \mathcal{M}(s_i)$ /* get existing prefixes that executes to state $s_i$ */
7:             **if not** $isDuplicate(\hat{y}_{\leq i}, Y_S)$ **then**
8:                 $\mathcal{B} \leftarrow updateBuffer(\hat{y}_{\leq i}, \mathcal{B})$
9:                 $\mathcal{M} \leftarrow updateMapping(\hat{y}_{\leq i}, \mathcal{M})$
10:             **end if**
11:             **continue** /* we only need the longest matching prefix */
12:         **end if**
13:     **end for**
14: **end for**
15:  $\theta \xleftarrow{\text{update}} \nabla_\theta \mathcal{L}(x, \mathcal{B}, P_\theta)$

---

of the known states for a PCS, *i.e.,* if $s_i \in \mathcal{M}$;
▷ **Function** *isDuplicate*$(\hat{y}_{\leq i}, Y_S)$: As mentioned in § 3.3.2, we use AST and length constraint to rule out "trivial variants" and identify new PCSs to save in the buffer $\mathcal{B}$. Here the solutions to compare are the set of solutions $Y_S$ that reaches the same intermediate state, *i.e.,* being state-based equivalent;
▷ **Function** *updateBuffer*$(\hat{y}_{\leq i}, \mathcal{B})$: Here we not only need to add the new PCS into the buffer $\mathcal{B}$, but also need to prune out the saved solutions that are prefix of $\hat{y}_{\leq i}$;
▷ **Function** *updateMapping*$(\hat{y}_{\leq i}, \mathcal{M})$: Here we need to save the states of all prefixes of an identified partially-correct solution, thus we will loop through all prefixes of $\hat{y}_{\leq i}$ and obtain its execution state, then update $\mathcal{M}$ accordingly. As mentioned above, existing PCSs may be a prefix of the new PCS, so we also need to prune out such existing PCSs from mapping $\mathcal{M}$.

## D   QUALITATIVE ANALYSIS

In Tab. 6, we show more examples of the fully-correct and partially-correct solutions that the models found during self-sampling, from both the MathQA and GSM datasets. First, we can see that for some NL problems, it is possible that no FCS or PCS can be found with self-sampling, as in *MathQA-Example-1* and *MathQA-Example-1*. Take *MathQA-Example-2* as an example, the question is quite straightforward thus it leaves very little room for the existence of other correct solutions, as the reference solution is already very short. Moreover, we can also observe that the ways self-sampled FCS and PCS differ from the reference solution vary a lot. In *MathQA-Example-2*, *GSM-Example-1* and *GSM-Example-2* the sampled FCSs complete the task with very different paths compared with the reference solution, and actually result in using fewer lines of code. Another way of getting FCS or PCS is to perform small and local perturbations, *e.g.,* switch the two sides of a addition or re-order the two non-dependent statements, as shown in other examples. We find that these local perturbations are more common in general in both datasets, as such patterns are easier for the model to learn.

## E   TRACKING TRAINING PROGRESS

**Learning from self-sampled solutions mitigates overfitting.**   Here we shown the PASS@$k$ performance curve with respect to the training process in Fig. 5. From the curves, we can observe that for MLE, while PASS@$1$ and PASS@$5$ generally improves during training, other PASS@$k$ for

| NL Problem Descriptions | Ref. Solution | Self-Sampled FCS | Self-Sampled PCS |
|---|---|---|---|
| *(MathQA-Example-1):* The charge for a single room at hotel P is 70 percent less than the charge for a single room at hotel R and 10 percent less than the charge for a single room at hotel G. The charge for a single room at hotel R is what percent greater than the charge for a single room at hotel G? | `n0=70.0`
`n1=10.0`
`t0=100.0-n0`
`t1=100.0-n1`
`t2=t0/t1`
`t3=t2*100.0`
`t4=100.0-t3`
`t5=t4/t3`
`answer=t5*100.0` | `n0=70.0`
`n1=10.0`
`t0=100.0-n1`
`t1=100.0-n0`
`t2=t0/t1`
`t3=t2*100.0`
`answer=t3-100.0` | - |
| *(MathQA-Example-2):* If john runs in the speed of 9 km/hr from his house, in what time will he reach the park which is 300m long from his house? | `n0=9.0`
`n1=300.0`
`t0=n0*1000.0`
`t1=n1/t0`
`answer=t1*60.0` | - | - |
| *(MathQA-Example-3):* A class consists of 15 biology students and 10 chemistry students. If you pick two students at the same time, what's the probability that one is maths and one is chemistry? | `n0=15.0`
`n1=10.0`
`t0=n0+n1`
`t1=n0/t0`
`t2=n1/t0`
`t3=t0-1.0`
`t4=n1/t3`
`t5=n0/t3`
`t6=t1*t4`
`t7=t5*t2`
`answer=t6+t7` | `n0=15.0`
`n1=10.0`
`t0=n0+n1`
`t1=n0/t0`
`t2=n1/t0`
`t3=t0-1.0`
`t4=n1/t3`
`t5=n0/t3`
`t6=t1*t4`
`t7=t5*t2`
`answer=t7+t6` | `n0=15.0`
`n1=10.0`
`t0=n0+n1`
`t1=n0/t0`
`t2=n1/t0`
`t3=t0-1.0`
`t4=n0/t3`
`t5=n1/t3` |
| *(GSM-Example-1):* Ellie has found an old bicycle in a field and thinks it just needs some oil to work well again. She needs 10ml of oil to fix each wheel and will need another 5ml of oil to fix the rest of the bike. How much oil does she need in total to fix the bike? | `n0=2`
`n1=10`
`n2=5`
`t0=n0*n1`
`answer=t0+n2` | `n0=10`
`n1=5`
`t0=n0+n1`
`answer=n0+t0` | `n0=10`
`n1=5`
`n2=2` |
| *(GSM-Example-2):* There is very little car traffic on Happy Street. During the week, most cars pass it on Tuesday - 25. On Monday, 20% less than on Tuesday, and on Wednesday, 2 more cars than on Monday. On Thursday and Friday, it is about 10 cars each day. On the weekend, traffic drops to 5 cars per day. How many cars travel down Happy Street from Monday through Sunday? | `n0=20`
`n1=100`
`n2=25`
`n3=2`
`n4=10`
`t0=n0/n1*n2`
`t1=n2-t0`
`t2=t1+n3`
`t3=n4*n3`
`t4=t0*n3`
`answer=t3+n2`
`\`
`+t2+t3+t4` | `n0=25`
`n1=2`
`n2=20`
`n3=100`
`n4=10`
`t0=n0-n1`
`t1=n2/n3*n0`
`t2=t0-t1`
`t3=t2+n4`
`t4=n0-t3`
`answer=t4+n3` | `n0=2`
`n1=25`
`n2=20`
`n3=100`
`n4=10` |

Table 6: More examples of self-sampled fully-correct (FCS) and partially-correct solutions (PCSs). "MathQA" denotes the MathQA-Python-Filtered dataset and "GSM" denotes the GSM5.5K-Python dataset. All solutions are from the *final buffer after training* a GPT-Neo 2.7B model, while learning from self-sampled FCS+PCS with the MLE-Aug loss.

higher $k$ actually decreases after reaching the peak performance in early epochs, which is consistent with previous findings (Cobbe et al., 2021). This is due to overfitting: in the early stage of training, the model is less confident about its predictions thus the sampled $k$ solutions are very diverse, and while training continues, it overfits to the one reference solution provided for learning thus leads to poor generalization when evaluated by PASS@$k$ with high $k$ values. Fig. 5 also shows

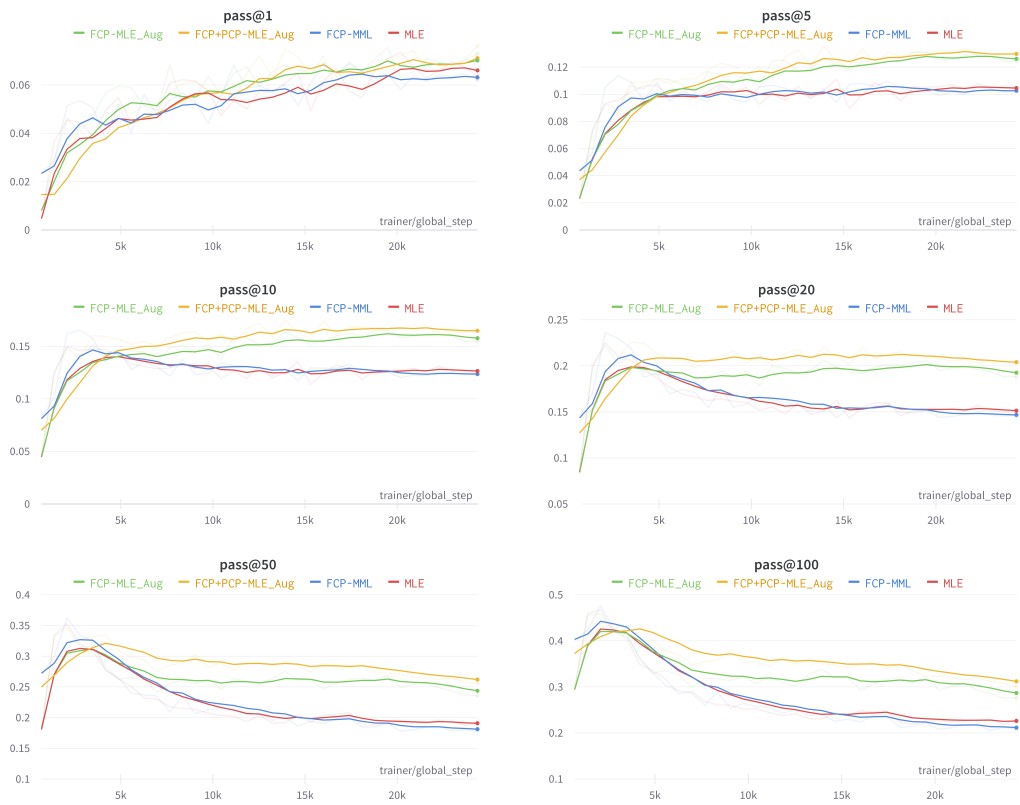

Figure 5: How PASS@$k$ on the dev set evolve during training. Results shown on GSM5.5K-Python dataset with GPT-Neo 125M model. Exponential moving average smoothing is applied for more clarity, but original curve is shown in shade.

how our proposed self-sampling method can mitigate the overfitting problem, as it keeps improving or maintaining PASS@$\{5, 10, 20\}$ while such performances start decreasing for MLE. Though it also shows improvements for PASS@$\{50, 100\}$, but the performance still decreases in later training stages. Here we can also see the importance of suitable learning objective, as MML has almost no effect in mitigating such overfitting issue.

**Early stopping is needed when prioritizing high $k$ value for PASS@$k$.** In our experiments, we select the model checkpoint with the best PASS@1 performance to evaluate all PASS@$k$. This setup aims to choose the best model that can solve the task with a small number of attempts (which corresponds to smaller $k$ value), as studied in (Austin et al., 2021). We can also observe that with our methods, the best PASS@1 checkpoint also yields the best or close to the best PASS@$\{5, 10, 20\}$ performances. However, in certain applications where large number of attempts are allowed, PASS@$k$ with high $k$ values should be prioritized. An example is to generate candidate solutions before reranking (Cobbe et al., 2021). In this case, an earlier checkpoint (*e.g.,* one with best PASS@100) should be used instead, which is not the best checkpoint for PASS@$k$ where $k$ is small. Also note that our proposed method are not suitable for these applications, as we observe no improvement on the peak PASS@$\{50, 100\}$ performances. We think this because when such peak performance is reached, it is still in the early stage of training thus not many FCSs or PCSs have been saved in the buffer yet.

**Partially-correct solutions help in early training stages.** To show how self-sampling effects training, in Fig. 6a we show how the size of the buffer progresses during training. From the curves, we can see that in the early training stages (*i.e.,* first 5k steps), the number of saved PCSs rapidly grows while the number of FCSs only slightly increases. In later stages of training, the growth of buffer size is mainly contributed by more FCSs being sampled and saved while the number of

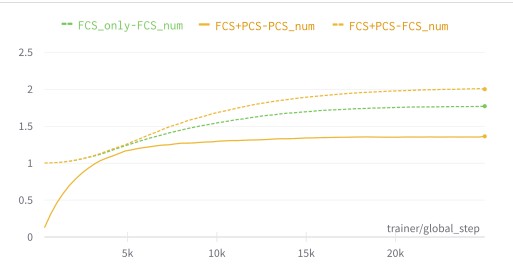
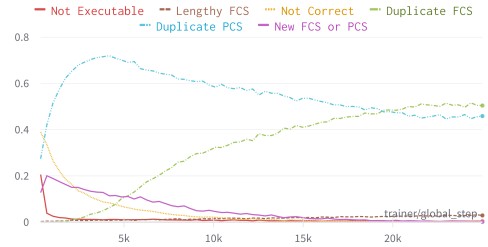

(a) Growth of the number of saved FCS and PCS during training. # FCS includes the gold solutions.

(b) Distribution of the characterization of self-sampled solutions during training.

Figure 6: How self-sampling evolves throughout the training process. Results shown as training the GPT-Neo 125M model on the GSM5.5K-Python dataset with MLE-Aug loss.

PCSs stays steady. Also when compared to learning only with FCSs, learning with FCSs + PCSs eventually accumulates more FCSs in the buffer (green dotted line vs yellow dotted line). In addition, we show how the distribution of the outcomes of self-sampled solutions changes throughout training in Fig. 6b. We can see that in the early training stages, the ratio of not executable/incorrect solutions quickly drops to almost zero. At the same time, the ratio of new FCS or PCS being saved reaches the peak. As training proceeds, the models are mostly sampling known FCS or PCS as the size of the buffer converges as well. But the number of self-sampled fully-correct solutions gradually overtakes the partially-correct ones.

