# OpenReview forum: "Learning Math Reasoning from Self-Sampled Correct and Partially-Correct Solutions"
_ICLR.cc/2023/Conference — ICLR 2023 poster_

### Official Review · Reviewer_bPyF · 2022-10-23

**Confidence:** 4
**Correctness:** 4
**Technical Novelty And Significance:** 3
**Empirical Novelty And Significance:** 3
**Recommendation:** 6

**Clarity, Quality, Novelty And Reproducibility:**

Overall, I found the paper to be well-written, although at times the writing could be improved, for example by adding additional motivation for the multi-target losses considered. The experimental evaluation is well-done and, given the model sizes involved, could be reproduced by others fairly easily. The main concern regarding reproducibility is the implementation of the solutions buffer, which involves lots of small details that would be difficult to get right without access to the original implementation; will it be released?

**Strength And Weaknesses:**

Overall, while the contributions are straightforward and, in the case of learning from partially-correct solutions, of somewhat limited scope (doesn’t generalise to more general coding tasks), the paper addresses important limitations of existing training methods for math reasoning tasks.

Regarding the proposed multi-target learning objectives, it’s not clear why MML would be expected to work. If multiple solutions produce the correct output, shouldn’t they all be rewarded? As in contrastive learning, wouldn’t we want to reward correct solutions, but downweight the probability mass assigned to incorrect ones? Additionally, I would have liked to see other objectives considered, such as ranking-based objectives or sequence-level (e.g. minimum bayes risk).

Regarding learning from partially-correct solutions, the additional filtering required (line 8 in Alg. 1) seems like an important limitation, given the footnote about the impact on performance. In general, identifying “trivial variants” seems like a hard problem, given the potential for malformed code during training which could make it hard to filter based on compiled ASTs.

In the limitations section, the issue of spurious solutions which achieve the correct solution by chance is raised. While it’s stated that this issue is “rare” in the domain under consideration, how was this quantified? Furthermore, given the persistent buffer of solutions maintained during training, is there a concern that early versions of the model may be more likely to produce spurious solutions, which would then remain throughout training in the solutions buffer?


**Summary Of The Paper:**

This paper considers the application of language models (LM) to mathematics reasoning tasks, noting that many existing datasets only contain one reference solution. This is problematic since many derivations may lead to the same answer, and therefore maximum-likelihood estimation of the LM may lead to overfitting. The main idea is the observation that samples generated from the model during training (“self-sampled”) can be evaluated to see if they yield the correct answer, and if so those alternate solutions may also be rewarded for a suitable choice of loss function. To this end, three loss functions are compared: MLE augmented with the additional samples (MLE-Aug) yielding the correct answer, maximum marginal likelihood (MML) which marginalizes over the candidate set, and smoothed MML which interpolates between MLE-Aug and MML.

Additionally, the paper proposes learning from partially-correct solutions. The challenge is to identify sequences which contain a prefix which evaluates to a necessary step of the correct solution. To do so, the idea is to exploit both self-sampled (correct) solutions and the gold reference to identify semantically equivalent prefixes (produce the exact same set of variable values). The longest partially-correct prefix is used for learning, which is distinguished from other sequences only by the lack of a distinguished end-of-sequence token.

Experiments on MathQA-Python and GSM5.5K-Python demonstrate that both self-sampled solutions and the inclusion of partially correct solutions improve model performance. The MLE-Aug loss function outperforms the MML losses in all cases. In addition, the resulting model produces diverse solutions, which likely contributes to the improved pass@k results.


**Summary Of The Review:**

While the scope of the paper is fairly limited (math reasoning), the contributions are worthwhile and soundly evaluated. There is a story (discussed in the limitations) for broader applicability of the proposed ideas. While the relatively small model sizes considered may limit the absolute performance of the model, this is also a strength when it comes to reproducibility, so in the balance I would say this is a positive.

---

> ### Author Response · Authors · 2022-11-17
> **Author Response to Reviewer bPyF**
>
> Thanks for the thorough review and the support for our work. Yes, we have released the code for review and we are also preparing for open source. To respond to your other questions and comments:
>
> * **The scope is somewhat limited**: please refer to the general response *"TO ALL REVIEWERS: further implications on the general text-to-code task"*
> * **Multi-target learning objective**: first we would like to clarify that we did *not* propose any new multi-target learning objective, as we were simply studying the commonly used objectives under the setting of learning from multiple self-sampled programs. For your concrete questions:
>   * **Why MML would be expected to work**: MML is a commonly used objective for weakly-supervised semantic parsing [1, 2, 3], which is quite related. It encourages the model to put all probability mass on all programs in B and no probability mass on any other programs. So MML does reward all the correct solutions in B, just not evenly, as shown in Tab 1.
>   * **Consider other objectives**: Thanks for the suggestions. In this work, we focused on exploring several common learning objectives that were used in the previous work, which of course do not cover all. But we are happy to discuss the implications of other loss functions:
>     * Ranking-based loss. We can think of two ways to use the ranking loss: 1) among all the sampled programs, optimize towards ranking the correct programs in front of the incorrect ones. This entails the fact that incorrect programs may still receive reward, which could make learning very noisy 2) among only the correct / partially-correct programs. This could be interesting while some heuristics (e.g., length) are needed to rank among the correct programs.
>     * Minimum bayes risk. This is interesting as we can see MBR as another way of relaxing the 0/1 binary correctness evaluation of programs, which is also the primary motivation for defining partial correctness in our work. However for MBR, we would also need a good risk function for the programs. Recent work such as [4] proposes to use execution consistency, and we are happy to explore how it can integrate with our method in the future.
> * **Difficulty in identifying trivial variants**: We don't have very complex protocols to prune the programs, as we simply use the length of the program with ASTs for pruning (by using the length of the gold program as reference). For malformed programs that can’t be parsed to AST, they are already pruned in the correctness filtering as in line 7 of Algorithm 1.
> * **The problem of spurious solutions**: This is a great question, and it is also typically referred to as the “spurious program” problem in semantic parsing. We would like to make three points here:
>   * In the experiments, we found that this issue rarely appears for math programming. To quantify it, we have manually looked into 100 examples from GSM5.5-Python and the corresponding self-sampled fully-correct solutions generated by the GPT-Neo 2.7B model, and we found that only 3/100 have spurious solutions. We believe this is because the correct answer is typically some decimal number which is improbable for a semantically wrong solution to obtain the correct answer by chance. We have also updated the limitation section of the draft to reflect the new results and discussions.
>   * We also considered this problem when formulating the definition of partial correctness, as we require the whole set of variables to match to establish program equivalence (see last paragraph of section 3.3.1).
>   * How to deal with spurious programs is an open problem for weakly-supervised semantic parsing (WSP) [1, 2, 3, 5]. However, different from WSP, in our case since we always have the gold reference program in the buffer for learning, it also balances out the effect of learning from spurious programs.
>
> [1] [Weakly-supervised Semantic Parsing with Abstract Examples](https://arxiv.org/abs/1711.05240)
> [2] [Neural Semantic Parsing with Type Constraints for Semi-Structured Tables](https://aclanthology.org/D17-1160)
> [3] [Iterative Search for Weakly Supervised Semantic Parsing](https://aclanthology.org/N19-1273)
> [4] [Natural Language to Code Translation with Execution](https://arxiv.org/abs/2204.11454)
> [5] [Inferring Logical Forms From Denotations](https://arxiv.org/abs/1606.06900)

---

> ### Author Response · Authors · 2022-11-29
> **A kind reminder**
>
> Thanks once again for your careful reviews and hope you find the responses helpful. Please let us know if there are any other questions that we can answer, as it is getting closer to the end of the discussion period.

---

### Official Review · Reviewer_dnGk · 2022-10-24

**Confidence:** 4
**Clarity, Quality, Novelty And Reproducibility:** The paper's presentation is clear and…
**Correctness:** 4
**Technical Novelty And Significance:** 3
**Empirical Novelty And Significance:** 3
**Recommendation:** 6

**Strength And Weaknesses:**

### strengths
In my opinion (authors please correct me if I'm wrong) the proposed approach seems like a clever data augmentation method. The technical contribution lies in finding partially correct solutions, which, thanks to the problem formulation, can be identified through the computation state of the generated solution (a code snippet/program). More broadly, I think this approach can be useful beyond MWP solving; for example, I can imagine applying similar approach to identify partially correct code/programs, which can potentially be useful for improving text-to-code models, with additional implications for computer science education for identifying and hinting a learner's partially correct code submissions.

### weaknesses
I have a question on the problem formulation. There is certainly nothing wrong with formulating the MWP solving as a text-to-code problem. My concern is: is this the best way to solve MWPs? Or put another way, is MWP the best setting to showcase the proposed method's effectiveness? In my opinion (correct me if I'm wrong), the trending approach for MWP solving specifically (I'm not talking about other math reasoning tasks) is to take advantage of models trained on natural language and generate solutions in natural language. This is because much of MWP's reasoning involves simple mathematics (often simple arithmetics, i.e., the two datasets that the authors considered) but a lot more commonsense reasoning and natural language narratives. I think there is a reason that recent methods take advantage of language model's capability in generating texts, and tackle MWP solving via chain-of-thought prompting, majority voting, etc., all of which rely on generating natural language texts that lead to the final solution.
- Could the author comment on why they take the text-to-code approach, rather than text-to-text approach, for MWP solving, and the pros and cons of each?
- Additionally, it would be helpful to compare these two approaches experimentally, although I understand it could be very laborious. Note that currently some of the better performing approaches for MWP solving can reach solution accuracy of > 50% [1, 2]

One more question: is it the case that the correctness criterion for a sampled solution only rely on whether the execution of the sampled solution gives the correct final answer? If this is the case, the fine-tuning data perhaps, and inevitably, will include samples whose final result is correct but intermediate computation step is wrong. How would this scenario impact the performance of the proposed approach?


[1] https://arxiv.org/abs/2110.14168

[2] https://arxiv.org/abs/2206.14858

**Summary Of The Paper:**

The paper proposes an approach to solve math word problems (MWPs) via fine-tuning with self-sampled solutions that can be partially correct. The approach relies on a text-to-code formulation: a generative model, which is to be fine-tuned, receives the MWP text and aims to generate a snippet of python code, which, upon execution, returns the solution to the input MWP. Under this formulation, the paper proposes to augment the training data with samples from the generative model itself. Samples whose computation returns the correct solution to the problem are marked as correct and are used as additional data for fine-tuning. Samples whose computation returns an incorrect solution but some intermediate steps are correct are marked as partially correct and are also used for fine-tuning. The paper establishes a method to find such partially correct solutions (via backtracking the implicit computation state of the generated code/program). Evaluation results suggest that fine-tuning with self-sampling and with partially correct solutions both contribute to improved performance.

**Summary Of The Review:**

The paper describes an interesting approach (text-to-code, self-sampling with partially correct solutions) to solve math word problems. I have some concerns on the problem formulation compared to the (seemingly more natural to me) text-to-text approach. I am very open to increase my score.

---

> ### Author Response · Authors · 2022-11-17
> **Author Response to Reviewer dnGk: Using text-to-text approaches instead of text-to-code**
>
> Thanks for the constructive feedback and we definitely hope that our work will have broader implications on improving text-to-code models and educational purposes.
>
> Regarding your question about *"using text-to-text approaches instead of text-to-code"*, we agree with the reviewer that text-to-text and text-to-code are two different approaches to solving MWPs and we do notice a recent trend of using prompting and the generation of natural language solutions to solve math reasoning problems. We want to emphasize that our goal in this paper is *not* to show that one approach is better than the other. There are advantages and disadvantages to both approaches as mentioned below and so, we believe it is important for the community to try to improve both approaches. Therefore, the goal of our work is to show how we can further improve a text-to-code approach for solving MWPs using self-sampled correct and partially-correct solutions.
>
> **Advantages of text-to-code approaches:**
> * Being able to use high-level APIs. As an example, some problems in MathQA would require calling APIs such as sqrt(), gcd(), etc. It is not clear how natural language solutions can handle it without execution of programs;
> * Faithful computation is always guaranteed as the generated programs are actually executed to obtain the final answer. Note that many current methods that generate text solutions still require a “calculator” component to fulfill this purpose [1, 2];
> * Text-to-code approach gives us the ability to identify other semantically correct programs and partially correct programs, which we leverage in this paper to further improve the model’s performance.
>
> **Advantages of text-to-text approaches:**
> * More interpretable and potentially cheaper to obtain annotations for training.
> * Better performance on large language models, especially used in combination of chain-of-thought prompting under few-shot settings.
>
> **Hybrid text-to-code,text approaches:**
> That said, we also believe that it is possible to combine the two approaches in the future so that we can get the best of both worlds. For e.g. the text-to-code approach can be augmented with inline natural language comments. In another ongoing work of ours, we found that using idiomatic code (i.e., program with grounded variable names, such as `num_candy_john`), codex-davinci-002 models achieves the performance of 65.7% on GSM8K, which is higher than both the vanilla chain-of-thought, and least-to-most prompting methods. We think this is because the programs intrinsically describe the program solving procedure. We believe that the technique presented in this paper about learning from self-sampled solutions would still apply to these hybrid approaches with some additional interesting research questions about how we can select solutions with good natural language descriptions/variable names for learning.

---

> ### Author Response · Authors · 2022-11-17
> **Author Reponse to Reviewer dnGk: all other questions**
>
> For your other questions:
>
> **Is MWPs the best way to showcase the effectiveness of the method**: We believe MWPs are perfect as a first step to study the definition and usage of partially-correct programs. The relatively simple variable types and program structures make it easy to trace the execution of the programs and establish program equivalence as shown in the paper. In this domain, we showed that partially-correct programs can also be useful for learning, which we hope to have further implications for more general programs as you mentioned.
>
> **Samples with correct final result but wrong intermediate steps**: This is a great question, and it is also typically referred to as the “spurious program” problem in semantic parsing. We would like to make three points here:
> * In the experiments, we found that this issue rarely appears for math programming. To quantify it, we have manually looked into 100 examples from GSM5.5-Python and the corresponding self-sampled fully-correct solutions generated by the GPT-Neo 2.7B model, and we found that only 3/100 have spurious solutions. We believe this is because the correct answer is typically some decimal number which is less likely for a semantically wrong solution to obtain the correct answer by chance. We have also updated the limitation section of the draft to reflect the new results and discussions.
> * We also considered this problem when formulating the definition of partial correctness, as we require the whole set of variables to match to establish program equivalence (see last paragraph of section 3.3.1).
> * How to deal with spurious programs is an open problem for weakly-supervised semantic parsing (WSP) [1, 2, 3]. However, different from WSP, in our case since we always have the gold reference program in the buffer for learning, it also balances out the effect of learning from spurious programs.
>
> [1] [Inferring Logical Forms From Denotations](https://arxiv.org/abs/1606.06900)
> [2] [Bridging Reinforcement Learning and Maximum Marginal Likelihood](https://arxiv.org/abs/1704.07926)
> [3] [Weakly-supervised Semantic Parsing with Abstract Examples](https://arxiv.org/abs/1711.05240)

---

> ### Author Response · Authors · 2022-11-29
> **A kind reminder**
>
> Thanks once again for your careful reviews and hope you find the responses helpful. Please let us know if there are any other questions that we can answer, as it is getting closer to the end of the discussion period.

---

> > ### Comment · Reviewer_dnGk · 2022-12-09
> > **Thank you**
> >
> > And apologies for a late response. I appreciate the authors' comments on the pros and cons of text-code vs. text-text (and hybrid) approaches for MWP and their insights on spurious solutions. I have increased my score, and hopefully the authors can examine a bit more of the sampled programs (currently the authors looked at 100) from diverse/different MWPs to could make their observation that only very few spurious programs appear more convincing.

---

### Official Review · Reviewer_2ima · 2022-10-25

**Confidence:** 4
**Correctness:** 3
**Technical Novelty And Significance:** 3
**Empirical Novelty And Significance:** 3
**Recommendation:** 5

**Clarity, Quality, Novelty And Reproducibility:**

Clarity: This paper is clearly written and well-structured. It is easy for the readers to capture the main idea.

Quality: The proposed method is technically sound and intuitive. The system built by the authors is potentially useful to the community and could encourage follow-up work on this topic.

Novelty: This paper does not present a brand new idea for tackling complex math reasoning problems, but it is a good example of combining several good ideas in the literature and making it work for an important task.

Reproducibility: I think the community may have trouble replicating the results if the code is not released as there are multiple stages for the proposed framework, e.g., online sampling and filtering, and each stage could potentially affect the final results.

**Strength And Weaknesses:**

Strengths:
- This paper successfully integrates several of the ideas in the literal, e.g., learning from execution, and semi-supervised semantic parsing, and improves the model performance on complex math reasoning.
- I liked the idea of using partially-correct solutions to encourage the model to explore the solution space. However, it would be better if the authors could explore a more general definition of the PCS.

Weaknesses:
- One of my major concerns about this paper is that the experiments were only conducted on two similar math reasoning datasets. Theoretically, the method could be generalized to other semantic parsing tasks like text-to-SQL and more general code synthesis. It would be more convincing if they can show the proposed method also works on other datasets/domains.
- I am also not totally convinced by the improvement presented in Sec 4.2, where the authors use Pass@k as the evaluation metric. We see that when $k$ is small, the improvement is quite marginal. It is quite possible that although the generated solutions are more diverse, the correct solution only appears a few times. I think using the majority vote of the generated $k$ solutions as the answer would be a better metric because, in the real use case, we only want to give the users a few solutions.

Other comments:
- One experiment I would like to see is *Number of saved FCSs and PCSs vs. Training iterations vs. dev performance*. When is the model able to sample diverse solutions? Do the diverse solutions decrease the convergence speed while improving the dev performance?
- In Table 2, the comparison to Codex is not fair as you can easily achieve much better performance with chain-of-thought prompting.
- I think the comparison of the different loss functions is not closely tied to the main idea of this paper. It would be better if the authors can use more space to explore the boundaries of the proposed method. For example, experiment with other semantic parsing tasks.

**Summary Of The Paper:**

Making pre-trained language models (PLMs) generalize to perform multi-step math reasoning problems is a challenging task. One of the main challenges is that during the fine-tuning stage, only one reference solution is used as supervision, preventing the model from exploring a more diverse solution space. To mitigate this issue, this paper proposed learning from self-sampled solutions. The self-sampling stage yields both fully-correct solutions and partially correct solutions, leading to more efficient exploration of the solution space. They conducted experiments on two math reasoning datasets, namely,  MathQA and GSM8K. The experimental results show that with training on FCS and PCS, the generated solutions are more diverse and improved the Pass@K over the baseline when $k$ is large. They also compared different loss functions and found that MLE-Aug loss function works the best when training with multiple solutions.

**Summary Of The Review:**

In general, this paper is well-motivated and contains several interesting ideas. However, I think the experiments are not convincing enough to show the generalizability of the proposed method (see weaknesses for details). At the current stage, I don't think it is not fully qualified as a top-tier conference publication.

---

> ### Author Response · Authors · 2022-11-17
> **Author Response to Reviewer 2ima**
>
> Thanks for the suggestions and sorry for the confusion, we hope to resolve your concerns by answering these questions:
> * **Only conducted experiments on math datasets**: please refer to the general response *"TO ALL REVIEWERS: further implications on the general text-to-code task"*
> * **Pass@k as the main evaluation metric**:
> pass@k is a common evaluation metric that is used to evaluate program generation tasks, including math reasoning problems [1, 3];
> though pass@1 is not generally improved by our method, improving pass@k also provides opportunities for reranking/clustering methods that leads potential improvements to pass@1 [2, 3, 4]
> To confirm that we are not only able to solve more tasks given k samples (i.e., higher pass@k), we added the evaluation of acc@k and to measure how often our methods produce *more* correct solutions in the k samples than baseline. On GSM5.5K, we found that while both approaches have acc@k=0 on 42.9% of the dev examples, our approach achieves better acc@k on 62.5% of the remaining examples (where at least one model has non zero accuracy).
> * **Comparison with Codex**: We agree that using chain-of-thoughts prompting might pose a stronger baseline for Codex, but note that we are using codex results more as a reference instead of a baseline to improve upon. Moreover, chain-of-thought is quite orthogonal to our contribution as we are mostly finetuning the models instead of prompting. But it could be interesting to see how it works in combination, such as using natural language descriptions as inline comments, etc.
> * **Comparison of loss functions**: We actually think the comparison of loss functions is one of our core contributions, as it addresses the problem of how to learn from the multiple targets, especially in the presence of a gold solution, self-sampled correct solutions, and self-sampled partially correct solutions, a combination that we believe has not been studied before. In fact, we initially thought MML would work well as it is commonly used in weakly-supervised semantic parsing [5, 6, 7], but eventually found it works poorly in our case. We gave a hypothesis in section 4.2 on page 7, as MML encourages the model to put all weights on one solution thus it has no advantage over MLE which is already learning from the gold program. We think other people in the community may learn from our results, which is also appreciated by several other reviewers.
>
> [1] [Program Synthesis with Large Language Models](https://arxiv.org/abs/2108.07732)
> [2] [Natural Language to Code Translation with Execution](https://arxiv.org/abs/2204.11454)
> [3] [Training Verifiers to Solve Math Word Problems](https://arxiv.org/abs/2110.14168)
> [4] [Competition-Level Code Generation with AlphaCode](https://arxiv.org/abs/2203.07814)
> [5] [Weakly-supervised Semantic Parsing with Abstract Examples](https://arxiv.org/abs/1711.05240)
> [6] [Neural Semantic Parsing with Type Constraints for Semi-Structured Tables](https://aclanthology.org/D17-1160)
> [7] [Iterative Search for Weakly Supervised Semantic Parsing](https://aclanthology.org/N19-1273)

---

> > ### Comment · Reviewer_2ima · 2022-11-18
> > **Reply to the author responses**
> >
> > Thanks for the clarification!
> >
> > Re: Pass@k as the main evaluation metric. I am not fully convinced that it is the appropriate evaluation metric to be used here even though it was used in past work. I briefly checked [1] and [3] and it seems they use pass@k more as an analysis of the model's different behaviors rather than the core improvement, correct me if I am wrong. Also, I still believe the performance of the majority vote is good to include.
> >
> > Re: Comparison of loss functions & Only conducted experiments on math datasets. Although I agree that modifying the proposed framework for other tasks may not be straightforward, I still think that the paper should put more weight on exploring at least **one different domain** rather than using that space for exploring different loss functions. As mentioned by the authors "MML encourages the model to put all weights on one solution thus it has no advantage over MLE", which is kind of intuitive to me as those sampled solutions are filtered to ensure high quality. With that said, I don't think it brings enough value to the community. On the other hand, showing the proposed framework can work on other tasks/domains can bring more interests to the community and provide a strong evidence that future work could be built upon the proposed one.

---

> > > ### Author Response · Authors · 2022-11-19
> > > **Reply to the followup questions**
> > >
> > > Thanks a lot for the quick response!
> > >
> > > **Majority vote as an evaluation metric**:
> > > We were actually waiting for our experiment to finish before we replied about the majority vote metric. We tested the majority vote results of 100 samples with our method vs. MLE on the GSM dataset. We found that while MLE's performance did not improve over pass@1 (i.e., stays at 7.4%), our method actually benefits from majority voting and improves from 7.5% to 8.3%. This result again confirms our hypothesis that improving pass@k and acc@k for higher k values is also important since we can then combine with methods like verification/clustering/majority voting to indirectly improve pass@1.  We will add these results to the next version of the paper.
> > >
> > > **loss function and different domains**
> > > Thanks for your comment. We want to reiterate that:
> > > * The focus of this paper is to improve the text-to-code methods for the math reasoning domain (which by itself is a very significant domain with several recent papers [1, 3, 8, 9]).
> > > * For this problem setting, we thoroughly investigated and presented our evaluation on how self-sampling can impact various metrics. The discussion of loss functions strongly ties with our method and our evaluation reveals several subtle observations about how different loss functions influence the different kinds of sampling strategies differently. For e.g. in Tab. 5, we can see that MML actually improves over MLE for pass@k if we only include FCS but the performance drastically drops when the model is learning from PCS as well. Also, we want to point out that our observation that MLE-Aug >> ($\beta$-)MML is in contradiction to the results in a well-known previous paper [10] because the settings are slightly different.
> > > * We are glad that all the reviewers think that our work (which was aimed at math domain) applies to general domains and we are excited to investigate that in future works.
> > >
> > > [8] [Solving Quantitative Reasoning Problems with Language Models](https://arxiv.org/abs/2206.14858)
> > > [9] [Chain of Thought Prompting Elicits Reasoning in Large Language Models](https://arxiv.org/abs/2201.11903)
> > > [10] [From Language to Programs: Bridging Reinforcement Learning and Maximum Marginal Likelihood](https://arxiv.org/abs/1704.07926)

---

> ### Author Response · Authors · 2022-12-09
> **A kind reminder**
>
> Thanks once again for your careful reviews and hope you find the responses helpful. Please let us know if there are any followup/other questions that we can answer, as there are only three days left in the discussion period.

---

### Official Review · Reviewer_bc3k · 2022-10-28

**Confidence:** 4
**Correctness:** 3
**Technical Novelty And Significance:** 3
**Empirical Novelty And Significance:** 2
**Recommendation:** 6

**Clarity, Quality, Novelty And Reproducibility:**

Clarity
- There is some lack of clarity around experiment details. The final training algorithm is a bit unclear around how many samples are generated, is the buffer refreshed whenever a problem is revisited.

Quality
- Overall the paper is well written,

Novelty
- The paper is related to a cluster of ideas around self-training, though the execution and the problem domain is somewhat new.

Reproducibility
- Code is not provided and experimental details are a bit lacking.

**Strength And Weaknesses:**

Strengths
- The idea is really interesting and there are some nice improvements in pass rates for k > 1.
- I liked the exploration of the different loss functions as they tend to weight self-generated samples differently.
- The approach to keep partially correct solutions is also interesting as it potentially allows for bigger solution set to finetune on.

Weaknesses
- The problem seems to be very code motivated, so it is not clear to me why more code datasets were not tried instead of math datasets (which are converted to code).
- pass@1 is not generally improved by this approach. This is a bit unsatisfying as, at test time, given a math problem one does not typically have test-cases to run the generated code against and verify the answer, so it is really pass@1 that matters more than any pass@k. Another concern is that maybe the pass@k just improves because there is more data (not necessarily correct) to finetune on that improves the diversity of model samples.
- Experiment details are not fully explained and it could be tough to reproduce.

Questions for authors:
- What if you were to just sample solutions and finetune on them without caring about correctness? Will this yield an improvement on pass@k as well or is it much worse?
- The supplementary says that only one solution is sampled per problem. This is a bit confusing. Are you saying that one sample is enough to get a correct solution from the model? Please clarify: (1) how many samples are there for each problem (2) how many problems are in a gradient update (3) how many solutions of those sampled execute to correct answer (as a function of training steps).

**Summary Of The Paper:**

The paper studies finetuning on model sampled solutions to coding problems. The idea is to sample python solutions to problems, keep solutions that return the correct result in a buffer and finetune on them. The authors experiment with a few different losses for finetuning on the self-sampled solutions and evaluate their approach on two math reasoning datasets. They find that while pass@1 is not always improved from this approach, pass@k (for k >1) shows decent improvement.

**Summary Of The Review:**

I am going with a weak accept. See weakness section for explanation. Experiments on more code generation problems (where pass@k is more reasonable metric) would have been more convincing, but I like the overall approach of this paper.

---

> ### Author Response · Authors · 2022-11-17
> **Author Response to Reviewer bc3k**
>
> Thanks a lot for the thorough review and support for our work! We agree that while we focus on the math reasoning domain, the ideas in our work could be applied to more general text-to-code tasks, for which we have included some discussion in the general response. We have also released the code for review purposes while preparing for open source.
>
> To answer more of your questions:
> * **Why more code datasets were not tried**: please refer to the general response *"TO ALL REVIEWERS: further implications on the general text-to-code task"*
> * **Using pass@k as evaluation metric**:
> pass@k is a common evaluation metric that is used to evaluate program generation tasks, including math reasoning problems [1, 3];
> though pass@1 is not generally improved by our method, improving pass@k also provides opportunities for reranking/clustering methods that leads potential improvements to pass@1 [2, 3, 4]
> To confirm that we are not only able to solve more tasks given k samples (i.e., higher pass@k), we added the evaluation of acc@k and to measure how often our methods produce *more* correct solutions in the k samples than baseline. On GSM5.5K, we found that while both approaches have acc@k=0 on 42.9% of the dev examples, our approach achieves better acc@k on 62.5% of the remaining examples (where at least one model has non zero accuracy).
> * **Learn from self-sampled programs regardless of correctness**: This is an interesting suggestion. We believe that such an approach might help in a “distilling” case, where the samples are from a much stronger model (e.g., codex) and we try to finetune a smaller model with those samples. However, ours is a “self-sampling” setting where we sample from and finetune the same model. In this setup, we suspect that without the additional correctness based filtering, the model will *not* learn any new information that it does *not* already know.
> * **Details about self-sampling**: Sorry for the confusion. We’ve initially included some details in Appendix A and we’ve updated it to make it clearer. In each training step, we sample 1 solution for each problem in the batch, with (effective) batch_size = 32. This means that loss per task is first computed with the loss function and the saved PCS and FCS in the buffer (details in section 3.2), then it is averaged across the 32 tasks for one gradient update.  We’ve also updated the draft to reflect those changes.
> * **Whether the buffer refreshes**: buffers are persistent and cumulative across training epochs, thus not refreshed. This is also described in the first paragraph of section 3.1
> * **Percentage of the self-sampled correct solutions during training**: We have added two new plots as Figure 6 (a) and (b) on page 17 to share more insights on how the results of self-sampling evolves during training, including the percentage of solutions being correct.
>
> [1] [Program Synthesis with Large Language Models](https://arxiv.org/abs/2108.07732)
> [2] [Natural Language to Code Translation with Execution](https://arxiv.org/abs/2204.11454)
> [3] [Training Verifiers to Solve Math Word Problems](https://arxiv.org/abs/2110.14168)
> [4] [Competition-Level Code Generation with AlphaCode](https://arxiv.org/abs/2203.07814)

---

> ### Author Response · Authors · 2022-11-30
> **A kind reminder**
>
> Thanks once again for your careful reviews. Please let us know if there are any other questions that we can answer, as it is getting closer to the end of the discussion period.

---

### Author Response · Authors · 2022-11-17
**TO ALL REVIEWERS: updates to the paper and code release**

Thanks a lot for the thorough review and insightful comments from all the reviewers! We have updated our draft to reflect the suggestions from the reviewers as well as clarifying several questions.

**Updates on the paper draft:**
* Added new experiment results and discussions on buffer size / characterization of self-sampled programs vs. training steps to reflect how the self-sampling process changes during training (briefly discussed in section 4.3 page 8, then with more details in appendix E page 16 (marked with “[NEW]” tag in red))
* Quantified the spuriousness of self-sampled solutions in our experiments (in section 5 page 9)
* Included more detailed hyperparameters we use for the self-sampling algorithm (in appendix A page 12)

**Reproducibility and code release:**
We attached the link to the anonymized codebase in a separate reply to all the reviewers and S/ACs for review purposes. We are also working on open-sourcing all the experiment code to the public for reproducibility and will attach a link to the GitHub repository when the anonymity period ends.

---

### Author Response · Authors · 2022-11-17
**TO ALL REVIEWERS: further implications on the general text-to-code task**

First we would like to thank reviewer bc3k, 2ima and bPyF for suggesting that our work has further implications for more text-to-code tasks beyond the math reasoning domain, and we definitely agree that in principle, our proposed methods can be applied to more program synthesis tasks. But we would also like to make the following remarks on why we choose to experiment on the math reasoning domain in this work:
* Math reasoning is an important and challenging domain that has received more and more attention lately [1, 2, 3, 4]. In this work, we explored how to finetune code language models to generate Python programs in solving math reasoning problems, which is a good complement of recent text-based methods mentioned by reviewer dnGk.
* Using math programming as a testbed, we wish to answer the research question of how the model can learn from self-sampled programs (by exploring different objective functions), and furthermore, investigate how partially correct solutions can be identified and used to improve the learning. We believe the insights we gained from the above explorations are valuable both in the math programming domain as well as other general domains mentioned by the reviewers.
* At the same time, we want to point out that applying our approach to general purpose code is not straightforward. There are several additional challenges that need to be solved and we feel that it is out-of-scope for a single paper to do justice to all the components. In particular, unlike Math programs which have simple structure and variable types, general purpose code has more complex control flow (e.g. if-else, for loops), which makes it harder for us to trace intermediate execution states. We discuss some potential ways to resolve these issues and make it applicable to general programs in section 5, but a full implementation and evaluation on a general code domain would dilute the contributions of this paper (especially, the effect of the learning objectives and the effect of partially correct programs). We will leave such extensions to future work. We are also excited to see how other people in the community would solve the above challenges for general purpose code (given that this paper established that self-learning and learning from partially correct programs helps).

[1] [Training Verifiers to Solve Math Word Problems](https://arxiv.org/abs/2110.14168)
[2] [Solving Quantitative Reasoning Problems with Language Models](https://arxiv.org/abs/2206.14858)
[3] [Program Synthesis with Large Language Models](https://arxiv.org/abs/2108.07732)
[4] [Chain of Thought Prompting Elicits Reasoning in Large Language Models](https://arxiv.org/abs/2201.11903)

---

### Decision · Program_Chairs · 2023-01-20

**Decision:**

Accept: poster

**Justification For Why Not Higher Score:**

I believe a poster is the right format to convey the empirical performance gain of the proposed sampling schemes. Details behind empirical results and experimental settings are best discussed in front of a small crowd in real-time.
There is not enough novel technical content for a spotlight.

**Justification For Why Not Lower Score:**

The paper is borderline, it could be rejected if the acceptance threshold needs to be re-calibrated.

**Metareview: Summary, Strengths And Weaknesses:**

The paper proposes to train large language models for math reasoning problems by augmenting the usual maximum likelihood estimation (MLE) scheme with self sampled instances. These techniques have been already explored in the literature of generative modeling. The main contribution is empirical: the authors evaluate the proposed augmented MLE on a two benchmarks for mathematical reasoning against vanilla baselines and other MLE variants such as marginal MLE and its weighted versions.

The reviewers convened that the problem addressed is relevant and the proposed self-sampling augmentation scheme can be potentially interesting. However, some skepticism around the statistical significance of the performance gains and the metrics employed has emerged during the reviewing and rebuttal phases.

Regarding the metric used, pass@K is indeed quite permissive for mathematical reasoning when K >> 1 and only acts as a proxy when evaluating these generative models. I, however, understand the authors when they say that they use the metric as it is quite common in this literature.

Concerning the significance of the gains, the augmented MLE scheme seems to generally yield improvements only when many possible output candidates are considered, e.g., when evaluating pass@K with K being 80 or more. I agree with reviewers that on GSM5.5K-Python pass@1, which is the arguably the metric of interest, is unaffected w.r.t. competitor baselines. However, we can see very significant improvements on the filtered version of MathQA-Python. This also tells a story about filtered vs non-filtered variants. Despite the lack of some statistical tests, I believe these results can still be considered as evidence in favor of using the proposed augmented MLE.

In summary, I believe the paper proposes a simple augmentation technique that can have a practical use also for mathematical reasoning applications.



**Note From Pc:**

if the above contains the word "oral" or "spotlight" please see: "oral" presentation means -> notable-top-5% and "spotlight" means -> notable-top-25%. As stated in our emails, we are disassociating presentation type from AC recommendations

**Summary Of Ac-Reviewer Meeting:**

Reviewers did not answer my emails, unfortunately. However, one reviewer increased their score post-rebuttal.